# The Impact of Physical Activity Restrictions on Health-Related Fitness in Children with Congenital Heart Disease

**DOI:** 10.3390/ijerph19084426

**Published:** 2022-04-07

**Authors:** Joel Blanchard, Brian W. McCrindle, Patricia E. Longmuir

**Affiliations:** 1Children’s Hospital of Eastern Ontario Research Institute, 401 Smyth Road, Ottawa, ON K1H 8L1, Canada; jblanchard@cheo.on.ca; 2School of Human Kinetics, University of Ottawa, Ottawa, ON K1N 6N5, Canada; 3The Hospital for Sick Children, 555 University Ave, Toronto, ON M5G 1X8, Canada; brian.mccrindle@sickkids.ca

**Keywords:** body mass index, strength, pediatrics, flexibility, gross motor skill, physical competence

## Abstract

Children with congenital heart disease (CHD) are often restricted from some types of physical activity (PA) despite the lack of evidence regarding the need to restrict recreational PA, apart from those with rhythm disorders. This study retrospectively investigated the associations between parent-reported activity restrictions (on-going need to restrict exertion, body contact or competition) and measures of health-related fitness among 236 children (8.2 ± 2.1 years, range 4–12 years) treated for single ventricle (*n* = 104), tetralogy of Fallot (*n* = 48), transposition of the great arteries (*n* = 47) or atrial septal defect (*n* = 37). Body mass index (BMI), moderate-to-vigorous physical activity (MVPA; 7 day accelerometry), strength, flexibility, and movement skill assessment results were collected from the baseline assessment research records for two studies completed in Ontario, Canada. A subset of 62 children also had physician-reported activity restrictions. Regression models empirically tested the goodness of fit between the dependent and independent variables. Participants with body contact restrictions from both parents and physicians had significantly higher BMI z-scores (0.23 ± 1.19 vs. −0.32 ± 0.85; t = 2.55; *p* = 0.04 and 0.66 ± 1.33 vs. −0.02 ± 0.98; t = 2.25; *p* = 0.02 for CDC and WHO scores, respectively). Otherwise, BMI z-score was not associated with patient variables (*p* > 0.36; sex, cardiac diagnosis, age, or activity restriction). Children with any type of parent-reported restriction (0.98 ± 2.06 vs. −0.08 ± 1.99; t = 3.77; *p* = 0.0002) were less flexible. Movement skill (TGMD-2) scores were 50% lower (25.1 ± 31.2 vs. 52.6 ± 28.6; F = 6.93; *p* = 0.009) among children with parent-reported competitive sport restrictions. Weekly MVPA (*p* > 0.18) and strength (*p* > 0.05) were not associated with activity restriction. Children whose parents reported PA restrictions were less flexible, and had decreased movement skill and increased BMI z-scores if the restrictions impacted competitive sport or body contact, respectively. Future research is recommended to confirm these results among larger samples of children who have both parent- and physician-specified PA restrictions.

## 1. Introduction

Obesity is associated with health risk factors such as high blood pressure, hypercholesterolemia and insulin resistance in children [1]. Health-related behaviors, such as daily physical activity (PA), combine with broader social, environmental, and biological “environments” to influence childhood obesity [2]. Children with obesity have decreased physical competence, as assessed by motor competence [3] and musculoskeletal fitness [4,5]. Obesity is equally common (25%) among children with congenital [6] and acquired heart disease, arrhythmia, and heart transplantation [7]. One research study suggested that children with CHD who are restricted from competitive sport are more likely to be overweight or obese [8], while another found no such relationship [9]. The prevention and treatment of obesity through PA and fitness are priorities for children with CHD in order to minimize their risk of cardiovascular and chronic disease [1] and motor impairment [10,11], as well as to maintain their overall health and fitness [5].

Children with congenital heart disease (CHD) are less active than healthy peers [12], with lower moderate-to-vigorous physical activity (MVPA) being associated with lower physical competence [13]. Cardiovascular function, and in particular maximal cardiorespiratory exercise capacity, has historically been suggested as the causative factor for the decreased physical activity and fitness of these patients. Data suggesting that physical activity levels are reduced even when maximal exercise capacity is within normal limits suggest the need to look beyond cardiovascular function to the broad range of correlates known to be associated with childhood physical activity. These include demographic and biologic (e.g., age and sex), psychological/emotional (e.g., PA intention and PA preference), behavioral (e.g., diet and previous PA) and sociocultural (e.g., parent modeling) factors [14]. Parents and caregivers play an important role in childhood development [15]. Studies have revealed that, regardless of disease severity, parents of children with CHD report higher levels of stress vigilance with their children than parents of healthy age-matched children [16]. With 41% of parents unable to accurately identify their children’s activity restrictions [17], and reports of activity restrictions differing between parents and physicians [18], uncertainty may prompt parents to create an environment for their children that reduces exposure to peers and/or PA, influencing their social competence and development [19]. There is no evidence regarding the need to restrict recreational PA among patients with CHD, apart from those with rhythm disorders [20]. However, PA restrictions remain common [21].

Given that inactive lifestyles are associated with an increased risk of obesity and cardiovascular disease, and that parents have a significant influence on the development of healthy lifestyle habits, the purpose of this study was to understand whether PA-restricted environments are associated with decreased measures of health-related fitness among children with CHD. The secondary purpose was to understand whether the impact of PA restrictions varied with the type of restriction. We hypothesized that (1) PA restrictions would be associated with higher BMI z-scores, lower daily MVPA and decreased physical competence; and (2) restriction of exertion would have the strongest effect in comparison to restrictions of body contact or competitive sport.

## 2. Materials and Methods

### 2.1. Study Design

Variations in BMI z-score, MVPA and physical competence among children with CHD were investigated using retrospective data obtained from research records in Toronto, Ontario, Canada. Body composition, daily MVPA and physical competence descriptive results were computed from baseline assessment results (i.e., prior to intervention). Patterns of association between various PA restrictions (as reported by physicians or parents) and measures of BMI z-score, daily PA and physical competence were examined. All study procedures were approved by the Research Ethics Board at The Hospital for Sick Children (protocol 1000012482) and conducted in accordance with the Tri-Council Policy Statement.

### 2.2. Population and Recruitment

Data were identified retrospectively for children, aged 4 to 12 years, who had one of four types of CHD (single ventricle, tetralogy of Fallot (TOF), transposition of the arteries (TGA) or atrial septal defect (ASD)) and who had completed measures of daily PA, physical competence and BMI z-score. Sex was determined from the medical chart of the participants. Subjects eligible for this study had previously participated in research on PA self-efficacy in children with CHD [13] or home-based rehabilitation in children with single-ventricle physiology after Fontan [10]. Cardiologist approval of study participation had been obtained for each patient prior to recruitment and written informed consent and assent had been obtained from parents and children, respectively. In the study on self-efficacy [13], the inclusion criteria were: (1) enrollment at least 1 year after repair of CHD, (2) 4 to 12 years of age at study enrollment, and (3) no genetic conditions or neurological or other impairments precluding the ability to complete assessments or restricting physical activity. Participants were only eligible for this study if they had a complete repair of an ASD (repair via transcatheter device closure or surgical repair), TGA (repair via arterial switch operation), or TOF (primary biventricular repair), or single-ventricle palliation via Fontan. Inclusion criteria for the study of home-based rehabilitation [10] were: (1) age 6 to 11 years at enrollment, and (2) at least 1 year post-Fontan palliation for single-ventricle physiology. The responsible cardiologist specified medically necessary activity restrictions. Children with disabilities that would limit full participation in the assessments or interventions were ineligible.

### 2.3. Physical Activity Restrictions

Physical activity restrictions were reported by the children’s physician and/or parents. Several types of PA restrictions were analyzed: (1) PA with exertion restrictions (must be able to rest or take breaks as needed), (2) PA with restriction from competitive sport, (3) PA with restrictions for activities involving body contact or (4) any PA restriction (including all of the preceding PA restrictions as well as other reported restrictions). In order to identify required PA restrictions, physicians completed a customized form by checking the boxes associated with the prescribed PA restriction(s). Parents were asked an open-ended question about the child’s activity restriction(s) and their responses were then classified into the same categories.

### 2.4. Body Composition

Height (cm) and body mass (kg) measures were extracted from the research data. BMI was calculated by dividing the body mass by the square of the height in meters (kg·m^−2^). World Health Organization (WHO) and Centers for Disease Control (CDC) BMI z-scores, which are adjusted for sex and age, were computed [22,23] to assess variations in body composition. As a normalized variable, z-scores were selected to eliminate some sources of variance in raw values and simplify clinical interpretation [24].

### 2.5. Physical Activity

Being valid in healthy children [25] and children with CHD [26], omni-directional accelerometers (Respironics Actical 2.1, Philips Healthcare, Andover, MA, USA) were used to measure daily minutes of MVPA [27]. The accelerometers were placed at the mid-axillary line above the iliac crest and recorded activity in 15 s epochs for five school days and 2 weekend days [28]. The cut-off point for MVPA was set at 1600 counts/min [25]. Data from 3 week days and 1 weekend day [28,29] with at least 8 h [10] or 10 h [13] of wear time were considered valid. To obtain a more accurate representation of participants’ level of PA, log sheets were given to all participants to record any unusual activities and activities performed while not wearing the accelerometer [28].

### 2.6. Physical Competence

For this retrospective study, physical competence was comprised of measures of movement skill and musculoskeletal fitness. Movement skills were evaluated using the Test of Gross Motor Development—Second Edition (TGMD-2), an assessment designed to test the gross locomotive (run, gallop, hop on one foot, standing jump, running leap, and sideways slide) and object control (overhand throw, batting, kicking, dribbling, catching, and rolling a ball) skills [30]. Standard scores for the object control and locomotive sub-scores and total TGMD-2 scores were calculated from validated reference values among a sample of 1208 school-aged children [30], to obtain scores independent of children’s age and sex. In this study, the TGMD-2, a test validated in children between 3 and 10 years old [30], was used to assess the movement skills of participants 4 to 12 years of age as it has been suggested as suitable for older children [31]. The TGMD-2 assessment can be administered relatively quickly (15–20 min) compared to other tools, has a standardized procedure that is easy to follow and is designed for administration by teachers and health professionals [30], factors that make it particularly interesting from a clinical point of view. The musculoskeletal fitness measures included measures of muscular strength and flexibility [29]. Muscular strength was measured with a hand-grip dynamometer (Takei Scientific Instruments, Niigata, Japan) by assessing each hand twice (alternating) and combining the maximum score for each hand (in kg). Flexibility was assessed with the sit-and-reach test [29]. The participants sat on the floor with their legs extended against a flexometer (Fit Systems Inc., Calgary, AB, Canada). The best of two attempts to stretch as far forward as possible without bending the knees was recorded to the nearest 0.1 cm. Handgrip strength and flexibility z-scores were computed from published data for Canadian children 6 to 12 years of age [29].

### 2.7. Statistical Analyses

There was no significant difference (*p* > 0.05) between the single-ventricle participants tested by Banks et al. [13] and those tested by Longmuir et al. [10] in BMI z-score, level of MVPA or physical competence (data not shown). Therefore, the data collected from both samples were combined for all analyses. A *p* value < 0.05 was considered statistically significant. A *p* value < 0.01 was considered statistically significant after the correction for multiple comparisons (Hochberg).

The associations among categorical variables were verified using the chi-square test. Pearson’s correlation coefficient was used to measure the statistical relationship between two continuous variables. Means between groups were compared using independent Student’s *t*-test for two-group comparisons or one-way ANOVA with post hoc Tukey’s honestly significant difference (HSD) for comparing more than two groups. Hochberg correction was used to avoid type 1 errors when all PA restrictions were compared at once. Bland–Altman analysis of the different BMI z-scores (CDC vs. WHO) showed that they were highly correlated, but with not negligible mean differences and measurement bias (Figure 1). These two measures of BMI were therefore analyzed as separate outcomes. Mixed linear modeling analyses were completed using SAS for Windows version 9.4 (SAS Institute Inc., Cary, NC, USA). Regression models were empirically tested to optimize the goodness of fit between the dependent and independent variables. The dependent variables of movement skill (locomotive, object, and total TGMD-2 scores), musculoskeletal fitness (handgrip and flexibility), MVPA and BMI z-score measurements were entered as outcomes in separate models. The independent variables, PA restrictions, CHD diagnosis and sex, were entered into each model as fixed effect predictors with age as a covariate.

## 3. Results

### 3.1. Participants

Data from a total of 236 children [101 (43%) girls] with TOF (*n* = 48), TGA (*n* = 47), ASD (*n* = 37) or single ventricle (*n* = 104) were included in the analyses. Table 1 summarizes the participants’ demographic and health-related fitness data. The mean age was 8.2 ± 2.1 years (range 4.0–12.7 years, median 8.3, IQR 3.5). The distribution by age group (4–6; 7–9; 10–12) did not differ by CHD group (chi-square = 6.35, *p* = 0.39). Although not different in age, children with single ventricle were significantly shorter (*p* < 0.05) than children with ASD. Mean body mass index z-scores (adjusted for age and sex) were 0.12 ± 1.30 (range −6.03 to 3.44) and −0.22 ± 1.24 (range −6.61 to 2.57) using WHO and CDC norms, respectively. Total movement skill (F = 3.08, *p* = 0.03) and grip strength (F = 5.58, *p* = 0.001) scores were higher among children with ASD compared to all other groups. Children with ASD or TGA were significantly more active on weekends (F = 4.39, *p* = 0.005) than children with TOF or single ventricle. Locomotor (F = 5.65, *p* = 0.02), object control (F = 20.17, *p* < 0.001), total movement skill (F = 15.69, *p* < 0.001) and flexibility (F = 281.23, *p* < 0.001) scores were higher among girls. Mean grip strength z-scores were higher among all CHD groups (ASD: 0.50 ± 1.12; TOF: 0.05 ± 0.95; TGA: 0.04 ± 1.08) in comparison to children with single ventricle (−0.27 ± 0.95). The increase in grip strength with age was higher in boys compared to girls (F = 6.09, *p* = 0.01). An interaction effect (F = 138.43, *p* < 0.001) was also noted between sex and age for the sit-and-reach z-score. Although boys and girls both become less flexible with increasing age, this decrease in flexibility is greater in boys (t = −9.75; *p* < 0.001) than in girls (F = 17.58; *p* < 0.001). Boys were more active on weekdays (F = 6.90, *p* = 0.009) and had higher total weekly physical activity (F = 5.00, *p* = 0.03). While parent-reported PA restriction information was available for 99% (234/236) of participants, only 26% (62/236) of participants had data from the responsible physician regarding exercise restrictions.

### 3.2. Impact of a Parent-Reported Restriction from Competitive Sport

Participants whose parents said they were restricted from competitive sports had significantly lower motor skill scores (Table 2, *p* < 0.05) than participants who were unrestricted, a pattern that was true for locomotive (t = 2.30, *p* = 0.02), object (t = 2.76, *p* = 0.006) and *TGMD-2* total standard scores (t = 2.90, *p* = 0.004). Results suggested that parent-reported restrictions from competitive sports had a larger impact on object and total *TGMD-2* standard scores among boys than girls, but the sample size of children with these restrictions was too small to statistically analyze that interaction effect. Muscular strength was significantly associated with CHD diagnosis and age when competitive sport restriction was included in the model (F = 0.68, *p* = 0.57).

### 3.3. Impact of a Parent-Reported Exertion Restriction

Parent-reported restrictions of the child’s level of exertion were not associated with any model variables (Table 3).

### 3.4. Impact of a Parent-Reported Restriction from Body Contact

Parent-reported body contact restrictions were not related to measures of health-related fitness (Table 4).

### 3.5. Impact of Any Type of Parent-Reported Physical Activity Restriction

Among children whose parents reported one or more activity restrictions of any type, object control skills were lower among children who had some type of activity restriction Table 5). Other motor skills and other health-related fitness variables were not associated with a physical activity restriction.

### 3.6. Impact of Any Type of Physician-Reported Physical Activity Restriction

The cardiologists for a subset of 74 children had indicated whether or not the children were to be restricted from one or more types of physical activities. Among this subset of children, girls had better locomotor (F = 5.27, *p* = 0.03), object control (F = 10.53, *p* = 0.002) and total (F = 10.80, *p* = 0.002) movement skills. Object control skills (F = 5.96, *p* = 0.02) and strength (F = 7.66, *p* = 0.007) increased with age. Flexibility was greater among girls (F = 119.70, *p* < 0.001), decreased with age (F = 13.10, *p* < 0.001) and was better among children with ASD or TOF (F = 6.88, *p* = 0.002). Girls had lower BMI z-scores (CDC: F = 6.59, *p* = 0.01; WHO: F = 8.38, *p* = 0.005). Physical activity minutes did not differ by model variables.

## 4. Discussion

In this retrospective study, the PA “environment”, as reflected in activity restrictions, was not associated with most measures of health-related fitness (strength, flexibility, BMI z-score, MVPA). Furthermore, the type of PA restriction had almost no effect on the movement skill of children with CHD. Any type of PA restriction was associated with decreased object control skills, and parent reports of restriction from competitive sport were associated with lower movement skill (locomotor, object control and total). In this research study sample, health-related fitness and PA participation reflected the patterns observed in population studies. Girls had a lower level of MVPA and muscular strength but better flexibility. The CHD diagnosis was also associated with the level of MVPA, children with single ventricle or TOF being less active than children with ASD and TGA. Single-ventricle patients were also weaker than all other CHD diagnostic groups. These findings suggest the need to investigate psychosocial and behavioral factors [14] as potentially important variables related to the decreased physical activity of these patients. These findings also align with recent research identifying the reduced muscle mass identified in young adults with a single-ventricle circulation [32].

### 4.1. Physical Activity Restriction and Body Composition

While the prevalence of obesity has increased in children with CHD [33], this population tends to have smaller body dimensions compared to healthy children [6]. The results suggesting that BMI z-score differences are not associated with parent or physician-reported PA restrictions align with results obtained among 143 children with anomalous aortic origin of the coronary arteries, 78 of whom had medical exercise restrictions [34]. These results contrast with results from a single-center study of 110 patients with various types of CHD [8], which found that children with PA restrictions had a higher BMI. They also contrast with a retrospective study of 172 older children (mean age 13.1 ± 2.9 years) with complex CHD (TOF, TGA, or single ventricle), which found that higher BMI was associated with parent reports of restriction to mild forms of exercise [35]. In that single-center study, restriction information was extracted from notes in the medical record, with no data on the type of PA restriction or parent-reported restrictions [8]. Since PA restrictions recorded in the medical record have been found to differ from those directly reported by physicians or parents [18], the results may reflect an unrecognized bias in the type of patients for whom physicians note activity requirements. The contradictory findings may also be explained by data for single-ventricle patients, which indicates that their body composition may lack skeletal muscle mass and have excess adiposity even when BMI is normal [32]. Further research among a large sample of children with various types of CHD and including both children and adolescents is recommended to clarify the impact of an environment that limits PA participation on BMI z-score. Such research should utilize objective and direct measures of PA participation, BMI z-score and activity restriction (parent and physician).

### 4.2. Physical Activity Restriction and Moderate-to-Vigorous Physical Activity

Children with CHD, such as those who have the Fontan procedure [36], are less active than the healthy population. However, approximately one third of children achieve the World Health Organization guideline [37] of at least 60 min of MVPA per day [38]. The 33% (77/236) of participants in this study who achieved that recommendation, aligns with previous reports among Canadian children without [39] and with CHD [38]. In the present study, the type of PA restriction was not related to objectively measured minutes of MVPA, whether activity minutes were measured during the week, on the weekend or over the full week. This finding contrasts with research among adolescents, where parent-reported PA restrictions were associated with self-reports of decreased vigorous exercise [35]. Adolescents with CHD also perform less vigorous intensity activity than peers [40], suggesting that the relationship between exercise restriction and behavior may differ by age.

### 4.3. Physical Activity Restriction and Movement Skill

Parent reports of children’s restriction from competitive sport were negatively associated with all measures of movement skill, with exertion restrictions associated with decreased object control skills. That restriction from competitive sport would have a significant association with movement skill development is surprising, given that “competitive sport” restrictions are intended to limit “organized, competitive, and skillful physical activities inside fixed rules of commitments and fair play, involving pressure to train or play, or continue to train/play, at a high intensity regardless of whether that intensity is desired by or recommended for the participant” [12]. The results of this study may, therefore, reflect misperceptions of the term “competitive sport” as referring to all activities at higher intensities or skill levels. It may be perceived that children are in a “competitive” environment when scores are kept or children decide to push themselves to win or be the best (i.e., high levels of exertion). These misperceptions may inappropriately restrict children with CHD from many of the activities that are essential for optimal development of their movement skills [12].

Contrary to what would be expected, especially for object control skills [41], girls with CHD had better movement skill standard scores than boys. These results may reflect the normative data used to calculate the standard scores, in which boys have better movement skills [30]. Based on the raw values (i.e., not standardized for age and sex), locomotive skills were higher in girls than boys (39.1 ± 7.0 and 37.5 ± 9.3, respectively, *p* = 0.23) and object control skills tended to be slightly higher in boys than girls (38.2 ± 8.2 and 37.2 ± 8.6, respectively, *p* = 0.39). The lower standard scores among boys indicates that they are much farther below their healthy peers, while girls with CHD have a lower standard to meet among healthy girls [41].

### 4.4. Physical Activity Restriction and Physical Fitness

Physicians’ [42] and parents’ [43] supportive attitudes towards PA are known to be associated with higher physical fitness in healthy children [44], suggesting that they create a positive, enabling PA environment. However, the lack of association between PA restrictions and BMI z-score or the level of MVPA makes us wonder whether PA restrictions really have a significant impact on the fitness of children with CHD. The weak associations between activity restrictions and measures of physical fitness observed in this study suggest further research is required. Study results indicating that, as children got older, they became stronger and less flexible were expected and similar to the literature for healthy children [29].

### 4.5. Understanding the Impact of Physical Activity Restriction

Although importance has been placed lately on the deleterious effect that exercise restriction could have on health and fitness [8,20], the results of this study suggest that age, sex and the CHD diagnosis likely have a stronger impact than most PA restrictions. The heterogeneity of results to date may be partially explained by the use of both objective [27] and subjective [35] measures of PA and limited data on the impact of activity intensity [35]. Differing sources of restriction data, including the medical chart [34], physicians [8] or parents [36] may also contribute to the heterogeneity of results, as PA restriction information from these sources is known to differ [18]. The lack of association between activity restriction and measures of BMI z-score, fitness, or habitual activity may reflect the diverse opinions as to what type of PA children with CHD could or could not do [12,20,45]. Regardless of specific restrictions from the physician, children with CHD may have a tendency to avoid certain types of activity since a consensus on appropriate activities is not yet clear. These results suggest that, despite the work performed to promote PA [20], additional efforts are still needed among children with CHD.

### 4.6. Study Strengths and Limitations

Trends in this study suggest that children who experience parent-reported activity restrictions obtained scores below those of healthy peers while those restricted by physicians are more similar to peers. Further research is required to investigate these trends because the relatively small number of children with activity restrictions in this study may have impacted the statistical power for detecting relationships with the physical competence measurements. These results emphasize the importance of collaboration and trust between parents and physicians in order to provide children with CHD with an optimal environment for the development of physical competence.

A large proportion of the physician PA restrictions were missing for children with TOF, TGA and ASD. It would therefore not be appropriate to generalize the physician restriction findings to all types of CHD, as these data were almost solely patients with single ventricle. These retrospective data were obtained from studies among children with CHD who volunteered for PA research, and who may be biased toward being more physically active than those who declined participation. All of the participants were followed in the same pediatric cardiology clinic. This increases the possibility of bias due to sampling and selection, affecting the external validity of this sample. However, this type of research is difficult to conduct, targeting a relatively rare population and requiring close cooperation between parents, physicians, and researchers.

Our sample size (*n* = 236) provided sufficient power to identify mean differences by sex [girls (*n* = 101); boys (*n* = 135)] or for comparing the mean of each CHD diagnosis [TOF (*n* = 47), TGA (*n* = 48), ASD (*n* = 37) or single ventricle (*n* = 104)]. However, statistical power was limited for comparing children with and without PA restrictions since the number of children with reported restrictions was quite small (Appendix A). Prospective research examining PA and physical competence among children matched for age, sex, CHD diagnosis and restriction is recommended.

In this study, no distinction was made between moderate and vigorous PA. The relationship between BMI and exercise intensity in healthy children is such that BMI decreases with activity of either moderate or vigorous intensity activity [46,47]. Since objectively measured PA does not differ by CHD severity [38], it would be interesting to investigate whether there is a threshold of PA intensity above which PA interacts with BMI z-score.

## 5. Conclusions

In this retrospective study of children with CHD, PA restrictions were not associated with BMI z-score, the level of MVPA or musculoskeletal fitness. Age, sex and the CHD diagnosis were more strongly associated with fitness and movement skill than most PA restrictions. Participants with parent-reported restrictions from competitive sport had lower movement skills and their movement skill scores were also lower than expected for age. Parent-reported exertion restrictions were also associated with lower object control skill scores. As with healthy children, it is important to promote environments that support PA and limit sedentary behavior among children with CHD, regardless of diagnosis. Increased attention is recommended to evaluate the association between PA restriction with fitness and movement skills among larger samples who have each type of PA restriction. Further research could also investigate the effect of activity intensity on BMI z-score in children with CHD by using both subjective and objective PA measures.

## Figures and Tables

**Figure 1 ijerph-19-04426-f001:**
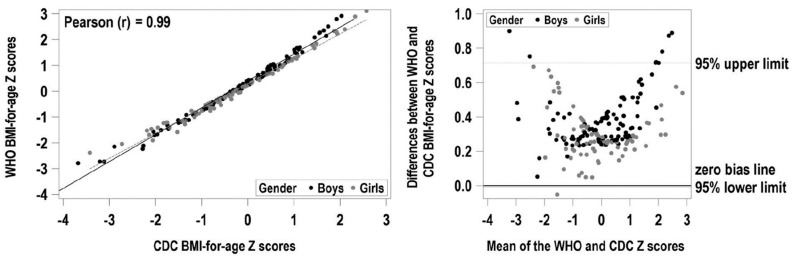
Relationship and agreement between World Health Organization (WHO) and Centers for Disease Control (CDC) body mass index z-scores. (**a**) Body mass index z-scores calculated using CDC norms. (**b**) Body mass index z-scores calculated using the WHO norms.

**Table 1 ijerph-19-04426-t001:** Participant characteristics.

Characteristics	ASD ^1^ (*n* = 37)	TOF ^1^ (*n* = 48)	TGA ^1^ (*n* = 47)	SV ^1^ (*n* = 104)	*p*-Value
** *Demographics* **					
No. ^1^ (%)	37 (15.7)	48 (20.3)	47 (19.9)	104 (44.1)	<0.001
Female no. (%)	21 (56.8)	23 (47.9)	14 (29.8)	43 (41.3)	0.08
Age, mean ± SD, yr	8.7 ± 2.1	8.1 ± 2.0	8.0 ± 2.2	8.2 ± 2.1	0.45
Age Group *n* (%)					0.39
4–6 yrs	8 (21.6)	16 (33.3)	20 (42.6)	33 (31.7)	
7–9 yrs	16 (43.2)	23 (47.9)	17 (36.2)	47 (45.2)	
10–12 yrs	13 (35.1)	9 (18.8)	10 (21.3)	24 (23.1)	
***Health-Related Fitness*** (mean ± SD)					
Locomotor	11.0 ± 1.8	10.0 ± 2.9	10.7 ± 2.7	9.8 ± 2.7	0.09
Object control	11.2 ± 2.3	9.8 ± 3.1	10.2 ± 2.8	9.5 ± 3.2	0.08
Total movement	22.2 ± 3.6	19.7 ± 5.4	20.9 ± 4.8	18.2 ± 5.1	0.03
Handgrip z-score	0.5 ± 1.1	0.05 ± 0.95	0.04 ± 1.1	−0.3 ± 0.9	0.001
Flexibility z-score	1.2 ± 2.3	1.1 ± 2.1	0.4 ± 1.8	0.3 ± 2.1	0.04
BMI ^1^ z-score (WHO ^1^)	0.08 ± 1.29	0.16 ± 1.43	0.03 ± 1.72	0.15 ± 1.10	0.93
BMI z-score (CDC ^1^)	−0.15 ± 1.29	−0.21 ± 1.39	−0.34 ± 1.61	−0.20 ± 1.00	0.97
***Physical Activity*** (mean ± SD)					
Weekday (mins),	62 ± 38	51 ± 21	63 ± 24	53 ± 21	0.03
Weekend (mins)	54 ± 45	37 ± 22	50 ± 31	38 ± 22	0.005 ^2^
Total weekly (mins)	424 ± 271	327 ± 141	416 ± 164	341 ± 138	0.008 ^2^

^1^ No: number of participants; ASD: atrial septal defect; TGA: transposition of the arteries; TOF: tetralogy of Fallot; SV: single ventricle; BMI: body mass index; CDC: Center for Disease Control and Prevention; WHO: World Health Organization; mins: minutes. ^2^ Children with ASD who were 7 to 9 years of age were significantly more active on weekends and in total weekly activity.

**Table 2 ijerph-19-04426-t002:** Association between health-related fitness and parent-reported restriction of competitive sport among children with CHD, adjusting for sex, age, and type of CHD.

Outcome	*n*	Restrict ^1^	Girls ^1^	ASD ^1^	TOF ^1^	TGA ^1^	Age	Int ^1^
**Physical Competence**								
Locomotive Std ^**1**^	201	−1.92 (0.95) ^2^	0.93 (0.38) ^2^	1.07 (0.6)	0.06 (0.5)	0.90 (0.5)	−0.06 (0.09)	8.13 (1.19) ^4^
Object Std	197	−2.40 (1.03) ^2^	1.76 (0.42) ^4^	1.35 (0.63) ^2^	0.05 (0.54)	0.87 (0.54)	0.11 (0.10)	5.65 (1.29) ^4^
TGMD-2 ^1^ Std	197	−4.27 (1.7) ^3^	2.66 (0.70) ^4^	2.52 (1.04) ^2^	0.21 (0.90)	1.86 (0.9) ^2^	0.04 (0.16)	13.77 (2.14) ^4^
Grip ^1^ z-scores	234	−0.41 (0.28)	−0.10 (0.12)	0.62 (0.17) ^4^	0.33 (0.2) ^2^	0.32 (0.2) ^2^	0.25 (0.03) ^4^	−2.62 (0.34) ^4^
S&R ^1^ z-scores	228	0.53 (0.25) ^2^	0.52 (0.22) ^2^	0.27 (0.23)	−0.31 (0.04) ^4^	1.18 (0.5) ^2^	−0.31 (0.04) ^4^	1.18 (0.50) ^2^
**Body Composition**								
BMI CDC ^1^	208	0.20 (0.44)	−0.07 (0.18)	0.05 (0.26)	−0.02 (0.24)	−0.14 (0.25)	0.02 (0.04)	−0.15 (0.53)
BMI WHO ^1^	191	−0.02 (0.49)	−0.18 (0.20)	−0.07 (0.28)	−0.001 (0.26)	−0.16 (0.28)	0.03 (0.05)	−0.03 (0.63)
**Physical Activity**								
Weekday ^1^	228	−8.23 (8.47)	−8.91 (3.36) ^3^	9.18 (4.81)	−1.71 (4.32)	8.20 (4.47)	1.30 (0.79)	38.1 (10.1) ^4^
Weekend ^1^	220	−15.05 (9.92)	−6.28 (4.02)	16.31 (5.80) ^3^	−1.31 (5.14)	9.72 (5.27)	−0.36 (0.93)	29.7 (11.9) ^3^
Weekly ^1^	219	−71.8 (58.9)	−55.2 (24.0) ^2^	82.7 (34.4) ^2^	−10.2 (30.5)	62.8 (31.6) ^2^	5.51 (5.57)	251.7 (71.0) ^4^

^1^ Restrict: children with parent-reported competitive sport restriction relative to those unrestricted; values reported as beta (SE). Girls: change in score relative to boys; CHD: change in score relative to children with single ventricle; ASD: atrial septal defect; TOF: tetralogy of Fallot; TGA: transposition of the arteries; Int: intercept; Std: standard score; TGMD-2: Test of Gross Motor Development version 2; Grip: handgrip; S&R: sit-and-reach; BMI CDC: body mass index z-score using Centers for Disease Control norms; BMI WHO: body mass index z-score using World Health Organization norms; Weekday: average daily minutes of moderate-to-vigorous physical activity on weekdays; Weekend: average daily minutes of moderate-to-vigorous physical activity on weekend days; Weekly: total minutes of moderate-to-vigorous physical activity for 5 weekdays and 2 weekend days. ^2^ Statistically significant (*p* < 0.05); ^3^ statistically significant (*p* < 0.01); ^4^ statistically significant (*p* < 0.001).

**Table 3 ijerph-19-04426-t003:** Association between health-related fitness and parent-reported restriction of level of exertion among children with CHD, adjusting for sex, age, and type of CHD.

		Predicted Values						
Outcome	*n*	Restrict ^1^	Girls ^1^	ASD ^1^	CHD TOF ^1^	TGA ^1^	Age	Int ^1^
**Physical Competence**								
Locomotive Std ^1^	201	−0.30 (0.69)	1.04 (0.39) ^3^	1.12 (0.59)	0.04 (0.50)	0.94 (0.51)	−0.06 (0.09)	9.59 (1.08) ^4^
Object Std	197	−0.04 (0.78)	1.88 (0.42) ^4^	1.47 (0.64) ^2^	0.05 (0.55)	0.95 (0.55)	0.11 (0.10)	7.83 (1.18) ^4^
TGMD-2 ^1^ Std	197	−0.17 (1.28)	2.88 (0.71) ^4^	2.71 (1.07) ^3^	0.20 (0.91)	2.00 (0.92) ^2^	0.05 (0.17)	17.55 (1.96) ^4^
Grip ^1^ z-scores	234	−0.19 (0.22)	−0.07 (0.12)	0.61 (0.17) ^4^	0.32 (0.15) ^2^	0.32 (0.15) ^2^	0.25 (0.03) ^4^	−2.45 (0.32) ^4^
S&R ^1^ z-scores	228	−0.23 (0.33)	3.11 (0.17) ^4^	0.52 (0.25) ^2^	0.51 (0.22) ^2^	0.27 (0.23)	−0.30 (0.04) ^4^	1.33 (0.48) ^3^
**Body Composition**								
BMI CDC ^1^	208	0.04 (0.33)	−0.09 (0.18)	0.04 (0.26)	−0.02 (0.24)	−0.15 (0.25)	0.02 (0.04)	−0.29 (0.49)
BMI WHO ^1^	191	0.01 (0.35)	−0.18 (0.20)	−0.07 (0.18)	−0.00 (0.26)	−0.16 (0.28)	0.03 (0.05)	−0.01 (0.57)
**Physical Activity**								
Weekday ^1^	228	−2.73 (6.19)	−8.35 (3.35) ^3^	9.24 (4.88)	−1.75 (4.35)	8.53 (4.48)	1.37 (0.79)	42.64 (9.24) ^4^
Weekend ^1^	220	−4.67 (7.45)	−5.30 (4.02)	16.51 (5.90) ^3^	−1.31 (5.19)	10.42 (5.29) ^2^	−0.22 (0.94)	38.2 (11.1) ^4^
Weekly ^1^	219	−31.6 (44.2)	−50.7 (23.9) ^2^	82.3 (35.0) ^2^	−10.9 (30.7)	65.3 (31.6) ^2^	6.25 (5.61)	282.9 (65.7) ^4^

^1^ Restrict: children with parent-reported competitive sport restriction relative to those unrestricted; values reported as beta (SE). Girls: change in score relative to boys; CHD: change in score relative to children with single ventricle; ASD: atrial septal defect; TGA: transposition of the arteries; TOF: tetralogy of Fallot; Int: intercept; Std: standard score; TGMD-2: Test of Gross Motor Development version 2; Grip: handgrip; S&R: sit-and-reach; BMI CDC: body mass index z-score using Centers for Disease Control norms; BMI WHO: body mass index z-score using World Health Organization norms; Weekday: average daily minutes of moderate-to-vigorous physical activity on weekdays; Weekend: average daily minutes of moderate-to-vigorous physical activity on weekend days; Weekly: total minutes of moderate-to-vigorous physical activity for 5 weekdays and 2 weekend days. ^2^ Statistically significant (*p* < 0.05); ^3^ statistically significant (*p* < 0.01); ^4^ statistically significant (*p* < 0.001).

**Table 4 ijerph-19-04426-t004:** Association between health-related fitness and parent-reported restriction of body contact among children with CHD, adjusting for sex, age, and type of CHD.

		Predicted Values						
Outcome	*n*	Restrict ^1^	Girls ^1^	ASD ^1^	CHD TOF ^1^	TGA ^1^	Age	Int ^1^
**Physical Competence**								
Locomotive Std ^1^	201	0.02 (0.48)	1.02 (0.39) ^3^	1.17 (0.60)	0.06 (0.51)	0.97 (0.51)	−0.06 (0.09)	9.91 (0.87) ^4^
Object Std	197	−0.46 (0.53)	1.84 (0.42) ^4^	1.32 (0.65) ^2^	−0.03 (0.55)	0.85 (0.56)	0.11 (0.10)	7.55 (0.95) ^4^
TGMD-2 ^1^ Std	197	−0.57 (0.88)	2.82 (0.71) ^4^	2.55 (1.09) ^2^	0.11 (0.92)	1.89 (0.93) ^2^	0.05 (0.16)	17.33 (1.58) ^4^
Grip ^1^ z-scores	234	−0.05 (0.15)	−0.09 (0.12)	0.63 (0.18) ^4^	0.33 (0.16) ^2^	0.33 (0.16) ^2^	0.25 (0.03) ^4^	−2.29 (0.26) ^4^
S&R ^1^ z-scores	228	−0.09 (0.22)	3.10 (0.17) ^4^	0.53 (0.26) ^2^	0.51 (0.23) ^2^	0.28 (0.23)	−0.30 (0.04) ^4^	1.49 (0.39) ^4^
**Body Composition**								
BMI CDC ^1^	208	−0.04 (0.22)	−0.09 (0.18)	0.02 (0.27)	−0.03 (0.24)	−0.17 (0.25)	0.02 (0.04)	−0.36 (0.41)
BMI WHO ^1^	191	−0.05 (0.24)	−0.18 (0.20)	−0.09 (0.29)	−0.01 (0.26)	−0.17 (0.28)	0.03 (0.05)	−0.04 (0.49)
**Physical Activity**								
Weekday ^1^	228	−2.10 (4.23)	−8.21 (3.37) ^3^	8.95 (4.99)	−1.97 (4.41)	8.41 (4.51)	1.36 (0.79)	43.9 (7.5) ^4^
Weekend ^1^	220	−1.55 (5.02)	−5.65 (4.03)	16.69 (6.05) ^3^	−1.29 (5.27)	10.59 (5.34) ^2^	−0.27 (0.94)	41.7 (8.96) ^4^
Weekly ^1^	219	−14.1 (29.7)	−52.8 (24.0) ^2^	82.2 (35.8) ^2^	−11.5 (31.2)	65.7 (31.9) ^2^	5.96 (5.59)	304.6 (53.2) ^4^

^1^ Restrict: children with parent-reported competitive sport restriction relative to those unrestricted; values reported as beta (SE). Girls: change in score relative to boys; CHD: change in score relative to children with single ventricle; ASD: atrial septal defect; TGA: transposition of the arteries; TOF: tetralogy of Fallot; Int: intercept; Std: standard score; TGMD-2: Test of Gross Motor Development version 2; Grip: handgrip; S&R: sit-and-reach; BMI CDC: body mass index z-score using Centers for Disease Control norms; BMI WHO: body mass index z-score using World Health Organization norms; Weekday: average daily minutes of moderate-to-vigorous physical activity on weekdays; Weekend: average daily minutes of moderate-to-vigorous physical activity on weekend days; Weekly: total minutes of moderate-to-vigorous physical activity for 5 weekdays and 2 weekend days. ^2^ Statistically significant (*p* < 0.05); ^3^ statistically significant (*p* < 0.01); ^4^ statistically significant (*p* < 0.001).

**Table 5 ijerph-19-04426-t005:** Association between health-related fitness and any type of parent-reported restriction among children with CHD, adjusting for sex, age, and type of CHD.

		Predicted Values						
Outcome	*n*	Restrict ^1^	Girls ^1^	ASD ^1^	CHD TOF ^1^	TGA ^1^	Age	Int ^1^
**Physical Competence**								
Locomotive Std ^1^	201	−0.19 (0.45)	1.01 (0.39) ^3^	1.05 (0.64)	−0.01 (0.53)	0.90 (0.53)	−0.05 (0.09)	9.76 (0.86) ^4^
Object Std	197	−1.16 (0.49) ^2^	1.83 (0.42) ^4^	0.78 (0.69)	−0.37 (0.57)	0.51 (0.57)	0.15 (0.10)	7.06 (0.92) ^4^
TGMD-2 ^1^ Std	197	−1.36 (0.81)	2.80 (0.70) ^4^	1.92 (1.15)	−0.28 (0.95)	1.49 (0.96)	0.10 (0.17)	16.78 (1.55) ^4^
Grip ^1^ z-scores	234	−0.15 (0.14)	−0.10 (0.12)	0.55 (0.19) ^3^	0.27 (0.16)	0.28 (0.16)	0.25 (0.03) ^4^	−2.36 (0.26) ^4^
S&R ^1^ z-scores	228	−0.35 (0.20)	3.07 (0.17) ^4^	0.34 (0.27)	0.39 (0.24)	0.16 (0.24)	−0.30 (0.04) ^4^	1.34 (0.37) ^4^
**Body Composition**								
BMI CDC ^1^	208	−0.02 (0.21)	−0.09 (0.18)	0.03 (0.29)	−0.03 (0.25)	−0.17 (0.26)	0.02 (0.04)	−0.34 (0.40)
BMI WHO ^1^	191	−0.04 (0.23)	−0.18 (0.20)	−0.10 (0.31)	−0.02 (0.28)	−0.18 (0.29)	0.03 (0.05)	−0.04 (0.48)
**Physical Activity**								
Weekday ^1^	228	0.12 (4.00)	−8.48 (3.35) ^3^	9.73 (5.37)	−1.48 (4.61)	8.87 (4.71)	1.33 (0.80)	45.4 (7.38) ^4^
Weekend ^1^	220	−0.47 (4.77)	−5.54 (4.02)	16.93 (6.51) ^3^	−1.15 (5.52)	10.71 (5.59)	−0.27 (0.95)	42.5 (8.79) ^4^
Weekly ^1^	219	−3.93 (28.26)	−51.7 (24.0) ^2^	84.6 (38.6) ^2^	−10.1 (32.7)	67.0 (33.3) ^2^	5.97 (5.65)	311.7 (52.2) ^4^

^1^ Restrict: children with parent-reported competitive sport restriction relative to those unrestricted; values reported as beta (SE). Girls: change in score relative to boys; CHD: change in score relative to children with single ventricle; ASD: atrial septal defect; TGA: transposition of the arteries; TOF: tetralogy of Fallot; Int: intercept; Std: standard score; TGMD-2: Test of Gross Motor Development version 2; Grip: handgrip; S&R: sit-and-reach; BMI CDC: body mass index z-score using Centers for Disease Control norms; BMI WHO: body mass index z-score using World Health Organization norms; Weekday: average daily minutes of moderate-to-vigorous physical activity on weekdays; Weekend: average daily minutes of moderate-to-vigorous physical activity on weekend days; Weekly: total minutes of moderate-to-vigorous physical activity for 5 weekdays and 2 weekend days. ^2^ Statistically significant (*p* < 0.05); ^3^ statistically significant (*p* < 0.01); ^4^ statistically significant (*p* < 0.001).

## Data Availability

To protect patient privacy, study data are available on request to the corresponding author.

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
