# Peer review of "The Impact of Physical Activity Restrictions on Health-Related Fitness in Children with Congenital Heart Disease"

_ijerph, 2022, doi:10.3390/ijerph19084426_

Round 1
Reviewer 1 Report
Dear Authors,
The article need some improvements:
Major issues.
The abstract should be clearer, how many studies? Subjects? Which are the restriction? 1-month restriction, one year, two years? Results are not well presented. Where are the conclusions?
Introduction must have justified much better the aims of the study, and to be more precise “bigger effect on body composition” it means on fat mass, bone mass, lean mass?
Discussion must be improved too, “locomotive skills were higher in girls than boys (39.1± 7.0 and 37.5 ± 9.3, respectively, p=0.23)” why? There is no significant differences.
Minor issues
To correct everything related with the abbreviations, in the main document and in the figure or tables, for example, they need to be explained the first time they come in the document, then they must be used, do not repeated the meaning. I also suggest to add some abbreviation like PA, the words physical activity is presented more than 50 times in the document, it will make the document shorter and easier to read.
Tables. What mean 1? Table 1, TOF n is 37 or 47?
To correct all contractions in the document, for example “among 234 children” “our sample size n = 236”
Best Regards
Author Response
|
Reviewer #1 Comment |
Author Response |
|
The abstract should be clearer, how many studies? Subjects? Which are the restriction? 1-month restriction, one year, two years? |
The abstract has been modified to indicate that date were extract from the records of baseline assessments completed during 2 studies. The typo in the number of participants has been correct (n=226). The parent-reported PA restrictions have been described as an “on-going need to restrict exertion, body contact or competition”), as all children were at least 1-year past their most recent surgery and therefore not subject to short-term or fluctuating restrictions. |
|
Abstract: Results are not well presented. |
The presentation of results in the abstract has been revised. Important associations are reported first, with non-significant findings at the end. |
|
Abstract: Where are the conclusions? |
The following concluding statement has been added to the abstract: “Children whose parents reported PA restrictions were less flexible, and had decreased movement skill and increased BMI z-scores if the restrictions impact competitive sport or body contact, respectively.” |
|
Introduction must have justified much better the aims of the study |
The introduction has been re-written to clarify the rationale and purpose of this research study. |
|
Introduction to be more precise “bigger effect on body composition” it means on fat mass, bone mass, lean mass? |
We have removed the term “body composition” from the manuscript. I has been replaced with the more accurate “body mass index z-score”, as it is only BMI that was available via the retrospective review of research records. It was not possible to obtain measures of fat mass, bone mass or lean mass from the research records. |
|
Discussion must be improved too, “locomotive skills were higher in girls than boys” Why? There is no significant differences. |
We apologize that the results were not clearly presented. Girls did have significantly higher movement skill scores for locomotive, object control and total TGMD-2 scores. We have clarified these findings in the results section and therefore have not changed the discussion of this point. |
|
To correct everything related with the abbreviations, in the main document and in the figure or tables, for example, they need to be explained the first time they come in the document, then they must be used, do not repeated the meaning. I also suggest to add some abbreviation like PA, the words physical activity is presented more than 50 times in the document, it will make the document shorter and easier to read. |
The abbreviations have been corrected. They are defined at each use and then used consistently throughout the remaining text. Explanations of the abbreviations have been retained in table and figure footnotes and the abstract. “Physical activity” has been replaced by “PA” as suggested by the reviewer. We have also replaced “moderate-to-vigorous physical activity” with “MVPA”. |
|
Tables. What mean 1? Table 1, TOF n is 37 or 47? |
The explanation of acronyms is now shown as footnote 1. The asterisk indicating a significant difference is now listed separately. The typo in table 1 (TOF=37) has been corrected. |
|
To correct all contractions in the document, for example “among 234 children” “our sample size n=236” |
We apologize for the inconsistencies in the originally submitted manuscript. We have reviewed the entire manuscript to ensure that the correct data are provided. |
Reviewer 2 Report
The authors retrospectively investigated the relationship between activity restrictions and health-related fitness measures in children with congenital heart disease. The authors address a topic with potential for strong societal impact bringing a glimpse of hope for parents desperately looking for any kind of help. The manuscript is very well written meeting all standards and I recommend its acceptation in present form (except for a typo at line 356: Impaact).

Author Response
|
Reviewer #2 Comment |
Author Response |
|
The authors retrospectively investigated the relationship between activity restrictions and health-related fitness measures in children with CHD. The authors address a topic with potential for strong societal impact bringing a glimpse of hope for parents desperately looking for any kind of help. The manuscript is very well written meeting all standards and I recommend its acceptation in present form (except for a typo at line 356: Impaact). |
Thank you for your kind comments about our research work. The typo that you identified has been corrected. |
Reviewer 3 Report
Dear Authors,
It is well known that physical activity is an important factor in the prevention of obesity as well as obesity-related diseases. Since children with congenital heart disease (CHD) have lower levels of both physical competence and physical activity, it is important to provide them with individualized support to maintain their health status. This study aimed to assess whether environments that restrict physical activity are associated with measures of health-related fitness among children with CHD (treated for single ventricle, tetralogy of Fallot, transposition of the great arteries or atrial septal defect). This article is undoubtedly of practical value.
The authors report on large sample of children, aged 4 to 12 years, who had CHD. However, no clear results are presented. In my opinion, the research methodology needs to be improved or better explained in order to solve the problem.
First, it is necessary to provide additional information about the health and neurological status of the participants. CHD often co-occurs with other disabilities, including neurological and physical (eg, congenital chromosomal syndromes), which can also affect and require restriction of physical activity.
Statistical data on the characteristics of patients (Table 1) need to be adjusted. The averaged parameters of age, height and body weight cannot be allowed, since children of a large age range were studied. The conclusion on the height of children with different CHD (lines 191-192) looks doubtful. The effects of height and age based on ANOVA statistics should be presented.
Second, authors hypothesized that physical activity restrictions play role in obesity in children with CHD. However, according to BMI parameters study group was quite normal. Thus, it is also necessary to discuss other reasons for motivation to exercise fitness in patients with CHD. It should be noted both in the Introduction and in the Discussion sections.
Thirdly, the statistical plan should be presented more clearly. The statistical criteria that were used to assess significance should be noted in the tables.
My conclusion is that the article is not ready for publication, as it has serious flaws, the analysis of the data was not carried out correctly, and the study was not conducted correctly.
Author Response
|
Reviewer #3 Comment |
Author Response |
|
The authors report on a large sample of children, aged 4 to 12 years, who had CHD. However, no clear results are presented. In my opinion, the research methodology needs to be improved or better explained in order to solve the problem. |
We thank the reviewer for this feedback. We have reviewed the manuscript and made edits throughout in the hopes of more clearly explaining the methodology used for this retrospective study and to address this general concern. We have also re-written the results section to more clearly present our data and analyses. |
|
First, it is necessary to provide additional information about the health and neurological status of the participants. CHD often co-occurs with other disabilities, including neurological and physical, which can also affect and require restriction of physical activity. |
We have clarified in 2.2 that the study participants did not have genetic, neurological or other impairments that would impact their study participation or physical activity. |
|
Statistical data on the characteristics of patients (Table 1) need to be adjusted. The averaged parameters of age, height and body weight cannot be allowed since children of a large age range were studied. The conclusion on the height of children with different CHD (lines 191-192) looks doubtful. The effects of height and age based on ANOVA statistics should be presented. |
We agree with the reviewer and have removed the averaged values for height and weight. The median and interquartile range have been added to describe the age of participants. Table 1 has been adjusted to indicate the number of boys and girls in 3 age groups (4-6, 7-9, 10-12 years). |
|
Second, authors hypothesized that physical activity restrictions play role in obesity in children with CHD. However, according to BMI parameters study groups was quite nomal. Thus, it is also necessary to discuss other reasons for motivation to exercise fitness in patients with CHD. It should be noted both in the Introduction and in the Discussion sections. |
Information has been added to the introduction discussing the broad range of physical activity correlates in children. We have also added similar information in the Discussion along with a study that suggests that the body composition of single ventricle patients may be abnormal (decreased skeletal muscle and increased adiposity) despite a normal BMI. |
|
Thirdly, the statistical plan should be presented more clearly. The statistical criteria that were used to assess significance should be noted in the tables. |
The description of the statistical plan (section 2.7) has been revised to more clearly describe the analyses completed. Footnotes are provided in each table indicating the criteria used to assess statistical significance. |
|
My conclusion is that the article is not ready for publication, as it has serious flaw, the analysis of the data was not carried out correctly, and the study was not conducted correctly. |
It is unfortunate that the reviewer perceived our manuscript in such a negative light. We believe that the study was conducted correctly and have revised the manuscript to more clearly convey the methods and results. In particular, the results section of the manuscript has been completely re-organized around the regression models that examined the impact of exercise restriction on health-related fitness variables, adjusting for sex, age and type of CHD. We hope that these changes will be satisfactory for the reviewer. |
Round 2
Reviewer 3 Report
Dear Authors,
The main purpose of this study was to understand if environments that restrict physical activity are associated with measures of health-related fitness (specifically body mass index (BMI) z-score, daily MVPA, or measures of physical competence (movement skill, strength, flexibility) among children with congenital heart disease (CHD).
Authors should clearly state the purpose of the study, which health problems in children with CHD can be prevented with adequate attitude and support for physical activity. The purpose and hypothesis of the study presented in lines 89-102 provide the same information. It is necessary to distinguish between these issues of research methodology.
The authors report a large sample of children aged 4 to 12 years with CAD. The manuscript retained some averaged data (for example, in the abstract and in tables), which cannot be allowed, since children of a large age range were studied.
Unfortunately, it is difficult to give an examination of the results due to the poor formatting of the tables.
My conclusion is that the article is not ready for publication, as it has serious flaws. The authors partially improved the manuscript, however, not all of the comments have been addressed properly.
Author Response
We thank the reviewer for providing additional feedback on the revisions to our manuscript. The following are the changes that we have made and our responses to the additional feedback.
1) Authors should clearly state the purpose of the study. Lines 89-102 provide the same information.
The last paragraph of the introduction has been revised. It now starts with the following sentence:
Given that inactive lifestyles are associated with an increased risk of obesity and cardiovascular disease, and that parents have a significant influence on the development of healthy lifestyle habits, the purpose of this study was to understand if PA-restricted environments are associated with decreased measures of health-related fitness among children with CHD.
2) The authors report a large sample of children aged 4 to 12 years. Averaged data in the abstract and tables cannot be allowed.
While we very much understand the reviewer's concerns, we believe that the averaged data reported in the manuscript is appropriate. In Table 1, we report both average age per diagnosis and data indicating that the distribution by age group does not differ by diagnosis. The variables for which average data are reported are z-scores or standard movement skill scores that are adjusted for the age and gender of the participants. We have also added the daily physical activity time by age group to Table 1. We believe that this makes it appropriate for our data to be represented as average values across the broad age range of our participants.
3) Tables are impossible to understand due to formatting.
We apologize to the reviewer that the tracked changes did not display correctly for the reviewer. We have "accepted" the tracked changes in all tables to ensure that they display as intended. We have also added information to Table 1 as described above.
